# BOUNDLESS SOCRATIC LEARNING

## ABSTRACT

An agent trained within a closed system can master any desired capability, as long as the following three conditions hold: (a) it receives sufficiently informative and aligned feedback, (b) its coverage of experience/data is broad enough, and (c) it has sufficient capacity and resource. In this position paper, we justify these conditions, and consider what limitations arise from (a) and (b) in closed systems, when assuming that (c) is not a bottleneck. Considering the special case of agents with matching input and output spaces (namely, language), we argue that such pure recursive self-improvement, dubbed '*Socratic learning*,' can boost performance vastly beyond what is present in its initial data or knowledge, and is only limited by time, as well as gradual misalignment concerns. Furthermore, we propose a constructive framework to implement it, based on the notion of *language games*.

## 1 INTRODUCTION

On the path between now and artificial superhuman intelligence (ASI; Morris et al., 2023; Grace et al., 2024) lies a tipping point, namely when the bulk of a system's improvement in capabilities is driven by *itself* instead of human sources of data, labels, or preferences (which can only scale so far). As yet, few systems exhibit such *recursive self-improvement*, so now is a prudent time to discuss and characterize what it is, and what it entails.

We focus on one end of the spectrum, the clearest but not the most practical one, namely pure self-contained settings of '*Socratic*' learning, closed systems without the option to collect new information from the external world. We articulate conditions, pitfalls and upper limits, as well as a concrete path towards building such systems, based on the notion of language games.

The central aim of this position paper is to clarify terminology and frame the discussion, with an emphasis on the long run. It is not to propose new algorithms, nor survey past literature; we pay no heed to near-term feasibility or constraints. We start with a flexible and general framing, and refine and instantiate these definitions over the course of the paper.

### DEFINITIONS

Consider a **closed system** (no inputs, no outputs) that evolves over time (see Figure 1 for an illustration). Within the system is an entity with inputs and outputs, called **agent**, that also changes over time. External to the system is an **observer** whose purpose is to assess the **performance** of the agent. If performance keeps increasing, we call this system-observer pair an **improvement process**.

The dynamics of this process are driven by both the agent and its surrounding system, but setting clear agent boundaries is required to make evaluation well-defined: in fact an agent *is* what can be unambiguously evaluated. Similarly, for separation of concerns, the observer is deliberately located outside of the system: As the system is closed, the observer's assessment cannot feed back into the system. Hence, the agent's learning feedback must come from system-internal **proxies** such as losses, reward functions, preference data, or critics.

The simplest type of performance metric is a *scalar* score that can be measured in finite time, that is, on (an aggregation of) episodic tasks. Mechanistically, the observer can measure performance in two ways, by *passively* observing the agent's behaviour within the system (if all pertinent tasks occur naturally), or by *copy-and-probe* evaluations where it confronts a cloned copy of the agent with interactive tasks of its choosing.

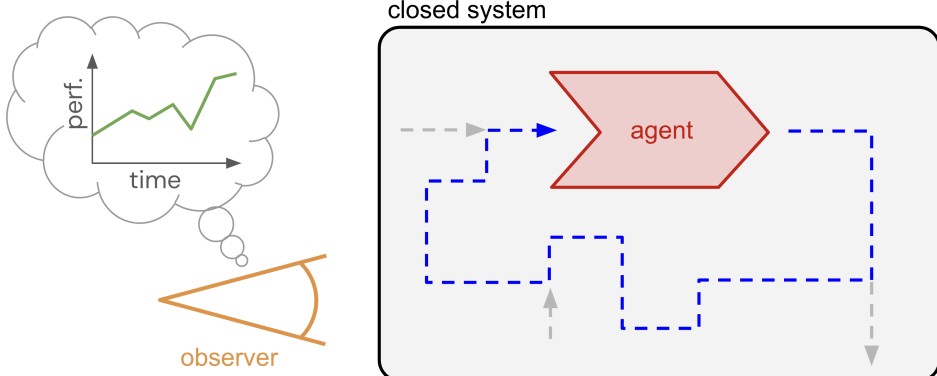

Figure 1: Cartoon depiction of the key definitions. An observer (**gold**) external to a closed system (**black**) assesses the performance (**green**) of an agent (**red**) over time. The process is one of *self-improvement* if agent outputs affect future agent inputs (i.e., some path like **blue** exists), and performance improves. Further, self-improvement is *recursive* if the agent input and output spaces are compatible, and the process is called '*Socratic learning*' if that space is language.

Without loss of generality, the elements within an agent can be partitioned into three types: *Fixed* elements are unaffected by learning, such as its substrate or unmodifiable code. *Transient* elements do not carry over between episodes, or across to evaluation (e.g., activations, the state of a random number generator). And finally *learned* elements (e.g., weights, parameters, knowledge) change based on a feedback signal, and their evolution maps to performance differences (Lu et al., 2023). We can distinguish improvement processes by their implied lifetime; some are *open-ended* and keep improving without limit (Hughes et al., 2024), while others converge onto their asymptotic performance after some finite time.[1]

## 2 THREE NECESSARY CONDITIONS FOR SELF-IMPROVEMENT

**Self-improvement** is an improvement process as defined above, but with the additional criterion that the agent's own outputs (actions) influence its future learning. In other words, systems in which agents shape (some of) their own experience stream, potentially enabling unbounded improvement in a closed system. This setting may look familiar to readers from the reinforcement learning community (RL; Sutton, 2018): RL agents' behaviour changes the data distribution it learns on, which in turn affects its behaviour policy, and so on. Another prototypical instance of a self-improvement process is *self-play*, where the system (often a symmetric game) slots the agent into the roles of both player and opponent, to generate an unlimited experience stream annotated with feedback (who won?) that provides direction for ever-increasing skill-learning.

From its connection to RL, we can derive necessary conditions for self-improvement to work, and help clarify some assumptions about the system. The first two conditions, feedback and coverage, are about feasibility in principle, the third (scale) is about practice.

### 2.1 FEEDBACK

Feedback is what gives direction to learning; without it, the process is merely one of self-modification. In a closed system where the true purpose resides in the external observer, but can not be accessed directly, feedback can only come from a proxy. This creates the fundamental challenge for system-internal feedback is be **aligned** with the observer, and remain aligned throughout the process. It places a significant burden on the system at set-up time, with the most common pitfall being a poorly designed critic or reward function that becomes exploitable over time, resulting in a process that deviates from the observer's intent. RL's famed capability for *self-correction* is not applicable here: what can self-correct is behaviour given feedback, but not feedback itself. Addi-

---

[1]Neither case needs to invoke a notion of optimality (Abel et al., 2024).

tionally, ideal feedback should be *efficient*, i.e., contain enough information (not too sparse, not too noisy, not too delayed) for learning to be feasible within the time horizon of the system.

## 2.2 COVERAGE

By definition, a self-improving agent determines the distribution of data it learns from. To prevent issues like collapse, drift, exploitation or overfitting, it needs to preserve[2] sufficient coverage of the data distribution everywhere the observer cares about. In most interesting cases, where performance includes a notion of generalisation, that target distribution is not given (the test tasks are withheld), so the system needs to be set up to intrinsically seek coverage, a sub-process classically called *exploration* (Ladosz et al., 2022). Note that aligned feedback is not enough for this on its own: even if a preferred behaviour is never ranked lower than a dispreferred one, that is not tantamount to guaranteeing that the agent will *find* the preferred behaviour.

## 2.3 SCALE

The research field of RL has produced a lot of detailed knowledge about how to train agents, which algorithms work in which circumstances, an abundance of neat tricks that address practical concerns, as well as theoretical results that characterize convergence, learning dynamics, rates of progress, etc. It would be futile to try and summarize such a broad body of work here. However, one general observation that matters for our argument is that 'RL works at scale': in other words, when scaling up experience and compute sufficiently, even relatively straightforward RL algorithms can solve problems previously thought out of reach (high-profile examples include: Tesauro et al., 1995; Mnih et al., 2015; Silver et al., 2016; 2018; Vinyals et al., 2019; AlphaProof & AlphaGeometry, 2024). For any specific, well-defined practical problem, the details matter (and differ), and greatly impact the efficiency of the learning dynamics; but the asymptotic outcome seems a foregone conclusion. The 'bitter lesson' of Sutton (2019) argues a related point: betting on scaling up computation (as opposed to building in human knowledge) has consistently paid off in the history of AI. Hence, with an availability of compute that keeps expanding, the resource constraints of agents (memory and compute) may be a transient concern; not all inefficiencies need to be fixed fully.[3]

## 3 SOCRATIC LEARNING

The specific type of self-improvement process we consider here is **recursive self-improvement**, where the agent's inputs and outputs are *compatible* (i.e., live in the same space), and outputs become future inputs.[4] This is more restrictive but less mediated than the general case where outputs merely influence the input distribution, most commonly instantiated by a (complex) *environment* that maps agent outputs into inputs. This type of recursion is an attribute of many open-ended processes, and open-ended improvement is arguably a central feature of ASI (see Hughes et al., 2024). On the other hand, compatibility is less restrictive than homoiconic self-modification, see Section 6.

An excellent example of such a compatible space of inputs and outputs is **language**. A vast range of human behaviours are mediated by, and well-expressed,[5] in language, especially in cognitive domains (which are definitionally part of ASI). As argued by Chalmers (2024) and a few centuries of rationalists before him (Cottingham, 1988), language may well be sufficient for thinking and understanding, and not require sensory grounding. Plus, language has the neat property of being a *soup of abstractions*, encoding many levels of the conceptual hierarchy in a shared space (see also Colas et al., 2022). A related feature of language is its extendability, i.e., it is possible to develop new

---

[2]This may entail conditions on how the system is initialised, as the agent needs to see a first set of inputs before it can produce its own.

[3]Not fully maybe, but learning needs to be efficient *enough* to take advantage of scale without saturating. A specific, timely tension here is around the role of the starting point of learning: some methods that attain mastery while learning purely from scratch (e.g., AlphaZero) while others start with broad competence (LLMs), but may not be as efficient in continuing to learn beyond that.

[4]Or at least some of them are fed back. Input and output spaces are not necessarily identical, but they intersect. For example, the agent could be generating code, but perceive natural language, (partly self-generated) code, and execution traces (Yang et al., 2023).

[5]"Whereof one cannot speak, thereof one must be silent." (Wittgenstein, 1921)

languages within an existing one, such as formal mathematics or programming languages that were first developed within natural language. While special-purpose tools (e.g., interpreters) for these are important for efficiency, natural language may be sufficient as a basis: just like humans can reason 'manually' through mathematical expressions when doing mental arithmetic, so can natural language agents (OpenAI et al., 2024). And of course, it does not hurt that AI competence on language domains has drastically improved recently, with a lot of momentum since the rise of LLMs. Early instances of LLM-mediated recursive self-improvement can be glimpsed in the meta-prompts of Fernando et al. (2023), the 'action programs' in Voyager's skill library Wang et al. (2023), and most recently, the self-reviewing, paper-generating 'AI scientist' (Lu et al., 2024).

For the remainder of the paper, we will use '**Socratic learning**' to refer to a recursive self-improvement process that operates in language space. The name is alluding to Socrates' approach of finding or refining knowledge through questioning dialogue and repeated language interactions, but, notably, without going out to collect observations in the real world—mirroring our emphasis on the system being closed. We encourage the reader to imagine an unbroken process of deliberation among a circle of philosophers, maybe starting with Socrates and his disciples, but expanding and continuing undisturbed for millennia: what cultural artifacts, what knowledge, what wisdom could such a process have produced by now?[6] And then, consider a question that seems paradoxical at first: In principle, how can a closed system produce open-ended improvement?

EXAMPLE

*To help make these ideas more concrete, we describe a hypothetical but not a priori implausible system (cf. Poesia et al., 2024). Consider the domain of mathematical statements (a subset of language).*[a] *The* **observer***'s performance metric is binary: has a proof for the Riemann hypothesis been found? The* **agent** *reads and writes mathematical statements and proofs (which are compatible input/output spaces). The* **system** *is closed, and contains the agent plus:*

- *a proof verifier (e.g., Lean)*
- *a collection $C$ of theorems or conjectures.*
- *a* **proxy** *reward for the agent: $+1$ for each verified new proof of a statement in $C$.*
- *a second collection $L$ of lemmas (or subgoals), initially empty.*

*The system allows the agent to produce proofs, verify them, formulate new statements, and add those to $L$. Over time, the agent may learn to simplify and decompose existing theorems, accumulate lemmas in $L$, learn to formulate lemmas that are more and more reusable, and increase the fraction of theorems in $C$ for which it can produce valid proofs. It self-improves. At some point, the expanding frontier of verified mathematical knowledge reaches a proof of the Riemann hypothesis, and the observer, satisfied, stops the system.*

[a] Note the restriction to a domain like mathematics, with verifiable feedback, is not fully representative of Socratic learning, as is sidesteps most of the challenge of feedback (Section 2.1).

## 4 The Fundamental Limits of Socratic Learning

Among the three necessary conditions for self-improvement, two of them, coverage and feedback apply to Socratic learning *in principle*, and remain irreducible. To make their implications as clear as possible, we ignore the third (the scale, practicality and efficiency concerns, see Section 2.3) in this section. We motivate this simplification by taking the long view: if compute and memory keep growing exponentially, scale constraints are but a temporary obstacle. If not, considering a resource-constrained scenario for Socratic learning (akin to studying bounded rationality) may still produce valid high-level insights.

The coverage condition implies that the Socratic learning system must keep generating (language) data, while preserving or expanding diversity over time. In the LLM age this has come to not seem too far-fetched: We can envision a generative agent initialized with a very broad internet-like

[6] To make this thought experiment compatible with our setting of a the single agent being evaluated, assume that the circle maintains the role of spokesperson, whose statements are judged by the observer, and who could be actively queried for evaluation.

distribution that produces a never-ending stream of novel language utterances. However preventing drift, collapse or just narrowing of the generative distribution in a recursive process may be highly non-trivial (Lewis et al., 2017; Shi et al., 2024).

The feedback condition requires the system to (a) continue producing feedback about (some subset of) the agent's outputs, which structurally requires a critic that can assess language, and (b) that feedback remains sufficiently aligned with the observer's evaluation metric (Christiano et al., 2018; Bai et al., 2022b). This is challenging for a number of reasons: Well-defined, grounded metrics in language space are often limited to narrow tasks, while more general-purpose mechanisms like AI-feedback are exploitable, especially so if the input distribution is permitted to shift. For example, none of the current LLM training paradigms have a feedback mechanism that is sufficient for Socratic learning. Next-token prediction loss is grounded, but insufficiently aligned with downstream usage, and unable to extrapolate beyond the training data. Human preferences are aligned by definition, but prevent learning in a closed system. Caching such preferences into a learned reward model makes it self-contained, but exploitable and potentially misaligned in the long-run, as well as weak on out-of-distribution data.

In other words, pure Socratic learning is possible, but it requires broad data generation with a robust and aligned critic. When those conditions hold, however, the ceiling of its potential improvement is only limited by the amount of resource applied. Current research has not established successful recipes for this yet, so the next section endeavours to make a concrete but quite general proposal for how to go about it.

## 5  LANGUAGE GAMES ARE ALL YOU NEED . . .

Fortunately, language, learning and grounding are well-studied topics. A particularly useful concept for us to draw on is Wittgenstein's notion of **language games**.[7] For him, it is not the words that capture meaning, but only the interactive nature of language can do so. To be concrete here, define a language game as an **interaction protocol** (a set of rules, expressible in code) that specifies the interaction of one or more agents ('players') that have language inputs and language outputs, plus a scalar **scoring function** for each player at the end of the game.[8]

Language games, thus defined, address the two primary needs of Socratic learning; namely, they provide a scalable mechanism for unbounded interactive data generation and self-play, while automatically providing an accompanying feedback signal (the score). In fact, they are the logical consequence of the coverage and feedback conditions, almost tautologically so: there is no form of interactive data generation with tractable feedback that is not a language game.[9] As a bonus, seeing the process as one of *game-play* immediately lets us import the potential of rich strategic diversity arising from multi-agent dynamics (as spelled out in depth in Leibo et al., 2019; Duéñez-Guzmán et al., 2023), which is likely to address at least part of the coverage condition. It also aligns with our intuition that dynamic, social co-construction (e.g., the circle of philosophers) has an edge over the self-talk of a single person that lives for millennia. Pragmatically too, games are a great way to get started, given the vast human track record of creating and honing a vast range of games and player skills (Berne, 1968); with Nguyen (2020) framing this richness as a demonstration of the fluidity of human agency and (local) motivations. Derrida might even argue that under the right lens, discourse is already structured as a game.[10] Colas et al. (2022) discuss a related set of ideas under the terminology of Vygotskian autotelic agents; while they do not assume a closed system, many of their 'internalised social interactions' could be cast as language games. A number of common LLM interaction paradigms are also well represented as language games, for example debate (Irving et al., 2018; Liang et al., 2023; Du et al., 2023), role-play (Vezhnevets et al., 2023), theory of mind (Kim et al., 2023), negotiation (Lewis et al., 2017; FAIR et al., 2022), jailbreak defense (Zeng et al.,

---

[7]"I shall also call the whole, consisting of language and the actions into which it is woven, the 'language-game'." (Wittgenstein, 1953)

[8]For simplicity, assume that games are guaranteed to terminate in finite time.

[9]Carse (2011)'s terminology is handy here too: we mean games of the 'finite' type that are played to win, as distinguished from 'infinite games' whose aim is to continue playing.

[10]"Every discourse, even a poetic or oracular sentence, carries with it a system of rules for producing analogous things and thus an outline of methodology." (Derrida, 1995).

2024), or outside of closed systems, paradigms like RL from human feedback (RLHF, Ouyang et al., 2022; Bai et al., 2022a; OpenAI et al., 2023).

### . . . IF YOU HAVE ENOUGH OF THEM . . .

Returning to our circle of deliberating philosophers: is there any *one* language game we could imagine them playing for millennia? Instead, maybe, they are more likely to escape a narrow outcome when playing **many** language games. It turns out that Wittgenstein (him again) proposed this same idea: he adamantly argued against language having a singular essence or function.[11]

Using many narrow but well-defined language games instead of a single universal one resolves a key dilemma: For each narrow game, a reliable score function (or critic) can be designed, whereas getting the single universal one right is more elusive (even if possible in principle, as argued by Silver et al., 2021).[12] From that lens, the full process of Socratic learning is then a *meta-game*, which schedules the language games that the agent plays and learns from (which is an 'infinite' game as per Carse (2011)). We posit that in principle, this idea is sufficient to address the issue of coverage (Section 2.2). Concretely, if a proxy of the observer's distribution of interest is available (e.g., a validation set of tasks), that can be used to drive exploration in the meta-game.

### . . . AND YOU PLAY THE RIGHT ONES

Socrates was famously sentenced to death and executed for 'corrupting the youth.' We can take this as a hint that a Socratic process is not guaranteed to remain aligned with external observers' intent. Language games as a mechanism do not side-step this either, but they arguably reduce the precision needed: instead of a critic that is aligned at the fine granularity of individual inputs and outputs, all that is needed is a 'meta-critic' that can judge which games should be played: maybe no individual language game is perfectly aligned, but what is doable is to filter the many games according to whether they make an overall net-positive contribution (when played and learned about). Furthermore, the usefulness of a game does not need to be assessed a priori, but can be judged post-hoc, after playing it for a while. Relatedly, a beneficial asymmetry is that it may be much easier to detect deviant emergent behaviour post-hoc than to design games that prevent it. All of these properties are forms of structural *leniency* that give the language games framework a vast potential to scale.

Stepping out of our assumption of the closed system for a moment: when we actually build ASI, we will almost surely want to not optimistically trust that alignment is preserved, but instead continually check the process as carefully as possible, and probably intervene and adjust the system throughout training. In that case, explicitly exposing the distribution of games (accompanied by interpretable game descriptions and per-game learning curves) as knobs to the designer may be a useful level of abstraction.

## 6 HIGHER-LEVEL RECURSIONS

So far, we discussed the minimal necessary form of recursion, a form of circularity that feeds (some of) the agent's outputs back to it. Within the framework of language games, two further types of recursion come to mind. The first idea is to tell the agent which game it is playing, and give it the choice to *switch* games, which game to switch to, and when to switch (Pislar et al., 2021). This is related to hierarchical or goal-conditioned RL, providing the agent with more autonomy and a more abstract action space. While shifting more responsibility into the agent, this setup could dramatically improve outcomes, as compared to a hardwired game-selection process outside of the agent—but of course this extra freedom could introduce additional risks of collapse or misalignment.

---

[11]"But how many kinds of sentence are there? Say assertion, question, and command?——There are *countless* kinds: countless different kinds of use of what we call 'symbols,' 'words,' 'sentences.' And this multiplicity is not something fixed, given once for all; but new types of language, new language-games, as we may say, come into existence, and others become obsolete and get forgotten." (Wittgenstein, 1953), emphasis in original.

[12]But, as a prescient Norbert Wiener was warning seven decades ago: "The machines will do what we ask them to do and not what we ought to ask them to do. [. . . ] We can be humble and live a good life with the aid of the machines, or we can be arrogant and die." (Wiener, 1949 / 2013).

Second, as games are interaction protocols that can be fully represented as code, they can live in a language agent's *output* space. Consequently, the agent could learn to **generate** games for itself to play.[13] Initially, it could simply produce local variations of exiting games, which adapt the difficulty level of theme, later on crafting recombinations of games, and ultimately ending up with *de novo* generation (Todd et al., 2024). This leads to second-order coverage concerns, in the space of language games instead of the space of language, to be addressed with filtering, prioritization, or curricula (Jaderberg et al., 2019; Parker-Holder et al., 2022).

The combination of both of these recursive extensions is an empowered agent that plays the full meta-game of how to improve itself via game generation and play. While appealingly elegant, this meta-game lacks the well-defined feedback mechanism of the inner language games, and it is an open research question whether established proxy metrics like learning progress would be sufficient to preserve both the coverage and alignment properties over time.

### SELF-REFERENTIAL SYSTEMS

The next and final step of recursion is recursive *self-modification*, that is, agents whose actions change their own internals, not merely influencing their input stream. These methods live on a spectrum characterized by the scope of what can be modified in such a way (and which elements remain fixed), and what amount of introspection, or access to its own workings, is available to the agent (Schaul & Schmidhuber, 2010). At the extreme end, a *fully self-referential* agent can observe and modify any[14] aspect of itself, without indirection. In principle, this type of agent has the highest capability ceiling; as asymptotic performance is capped by its fixed structure, unfreezing some of it and making it modifiable can only increase that upper bound—in particular, it is always possible to set the newly-flexible parameters to how they were while frozen, to recover the performance of the less-flexible agent (modulo learning dynamics that could get into the way). Past proposals for how to design self-referential systems were not (intended to be) practical (e.g., Schmidhuber, 1993; 2003; Schmidhuber et al., 1997; Kirsch & Schmidhuber, 2022), but modern LLMs' competence in code comprehension and generation is changing the playing field and may soon move these ideas from esoteric to critical.

## 7 CONCLUSION: OPEN-ENDED SOCRATIC LEARNING IS POSSIBLE

We set out to investigate how far recursive self-improvement in a closed system can take us on the path to AGI, and are now ready to conclude on an optimistic note. In principle, the potential of Socratic learning is high, and the challenges we identified (feedback and coverage) are well known. The framework of language games provides a constructive starting point that addresses both, and helps clarify how a practical research agenda could look like. We leave the fleshing out of that roadmap to future work, but the overall direction is becoming apparent. In particular, an understudied dimension is the breadth and richness of the *many* such language games. We think a great place to start is with processes capable of open-ended game generation. And not without seeing the irony, we propose all these ideas to scrutiny within an academic setting instead of resorting to self-talk in a closed system.

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
