# OpenReview forum: "Boundless Socratic Learning"
_ICLR.cc/2025/Conference — Submitted to ICLR 2025_

### Official Review · Reviewer_BsWX · 2024-10-21

**Soundness:** 2
**Presentation:** 3
**Contribution:** 1
**Rating:** 3
**Confidence:** 4

**Summary:**

The paper outlines a very broad abstract framework, at a philosophical level, for how self-improving AI without an upper bound in capabilities (i.e., scaling toward superintelligence) could be created. The main restriction of their framework is that of a closed system without input from the outside world. Within the closed system, arbitrary interactions of the agent with the system are possible. Specifically, the authors discuss socratic learning (outputs become future inputs), and specifically language games, where several agents compete against each other to achieve a high score. The authors argue that within this broad framework, recursive self-improvement to superintelligence is in principle possible, and wish to provide a framework of concepts to ground the discussion of other researchers about these topics.

**Strengths:**

The main strength is that this paper grapples at all with questions of the development of artificial superintelligence and recursive self-improvement, which are severely underdiscussed in the academic literature compared to their importance and plausibility.
I also think it is important to think about the conditions that allow for recursive self-improvement to get a better handle on how things are (or should be) developing, and this paper tries to be a step in this direction.

**Weaknesses:**

The paper does not propose theoretical or empirical results, and so it is essentially philosophy, and could as such maybe be considered a position paper. (The authors themselves call it a whitepaper). This means that in order to evaluate this work, we need to judge its *thinking* relative to prior work: in what sense does this paper enhance the reader's ability to think clearly about issues of superintelligence and recursive self-improvement, relative to the body of literature that already exists?

Unfortunately, I think this paper is not very grounded in the existing literature (more on that especially in point 2 below), which means I need to evaluate the philosophical thinking in this paper directly. However, when doing this, I immediately start to disagree with aspects of this paper, especially related to the "closedness of the system" that's part of their framework (see point 1 below). In disagreeing, I am myself bound to "philosophize", which means that my review might not appear very grounded in literature either; I hope this is excused given the nature of this paper. Overall, I thus think that while this paper engages with an important topic, it feels not ready for publication.

I recommend the authors to look at the paper [The Alignment Problem from a Deep Learning Perspective](https://openreview.net/forum?id=fh8EYKFKns) for an example of a position paper that was published at ICLR despite being pure philosophy; what made this paper worthwhile is partially that it engaged very extensively with prior work, thus providing a much more stable base for future engagement.

# 1. On Closedness of the system
The authors argue that it is in principle possible to create a recursively self-improving superintelligence in a closed system, i.e., a system that does not interact (via outputs or inputs) with the outside world. Language games are then their main instantiation of this procedure. I disagree with this, for the following reasons:
- Eventually, any true superintelligence will have to act in the real world to create new knowledge, e.g. by steering experiments. This likely requires to already interact with the world in the learning process itself since any closed system we could encode today would miss information about the world that we did not yet discover.
- Likewise, I think alignment (with society/preferences/humans etc.) might be impossible to achieve in a closed system. To solve this, the authors propose to have a meta-critic that evaluates which games should be played to lead to an aligned system; however, no reasons are given for how to design this meta-critic to achieve alignment. The authors seem themselves not fully convinced, since in line 259 they write "Stepping out of our assumption of the closed system for a moment: when we actually build ASI, we will almost surely want to not optimistically trust that alignment is preserved, but instead continually check the process as carefully as possible, and probably intervene and adjust throughout the training process." However, if closedness is an optional part of their framework, then it becomes an "everything framework" without bounds, thus making it hard to allow an informed and concrete discussion.
- I actually think recursive self-improvement will, at least in practice, not happen in a closed system. We already have the example of [AlphaChip](https://deepmind.google/discover/blog/how-alphachip-transformed-computer-chip-design/), an AI that feeds into better chip design (and thus more compute, which enables better future AI). Similarly, there are companies that try to [steer toward AI that can do ML research](https://openai.com/index/mle-bench/). Arguably, the AI would not act in a closed system but train other (better) AI systems on real-world servers, initially in collaboration with humans.
- I think one can also make the meta argument that researchers have a bad track record of proposing games or task taxonomies that comprise "all there is to learning". E.g., in the past, people thought once we have a chess AI, we will have solved AGI. Now, some people think once we have superhuman math-AI, we will have AGI. This paper seems to suggest that once we have sufficiently good learning procedures in closed language games, we will have AGI. I am a priori skeptical of such claims unless they are very well supported and address possible limitations / counter-arguments / existing literature etc.


# 2. Missing engagement with other work
There is a lot of other work that engages with the question of how (aligned) superhuman AI could or will be developed, and the challenges that arise therein. This paper does not demonstrate engagement with that work. A non-comprehensive list of works that could be engaged with (Very steered toward my personal interests; I am sure there are many others that make sense or that I could name if I'd thought about it for longer):
- Iterated distillation and amplification, see [here](https://arxiv.org/abs/1810.08575) and [here](https://ai-alignment.com/iterated-distillation-and-amplification-157debfd1616).
- The original [debate proposal](https://arxiv.org/abs/1805.00899).
- [11 proposals for building safe advanced AI](https://arxiv.org/abs/2012.07532).
- [constitutional AI](https://arxiv.org/abs/2212.08073), a method that could actually make closedness of the system somewhat easier for alignment (offering a different perspective from my perspective above that alignment isn't possible in a closed system).
- [Coherent extrapolated volition](https://intelligence.org/files/CEV.pdf). This relates to the following paragraph from the paper: "We encourage the reader to imagine an unbroken process of deliberation among a circle of philosophers, maybe starting with Socrates and his disciples, but expanding and continuing undisturbed for millennia: what cultural artifacts, what knowledge, what wisdom could such a process have produced by now?"


# 3. Terminology:
The terminology of recursive self-improvement isn't well thought-through. The definition in the paper seems to be this sentence:
> "The specific type of self-improvement process we consider here is recursive self-improvement, where the agent’s inputs and outputs are compatible (i.e., live in the same space), and outputs become future inputs."

However, the *idea* of recursive self-improvement is of course that the system improves itself, which is not part of this definition. Additionally, we can consider ways of recursive self-improvement that do not feed outputs into future inputs (e.g., if the AI rewrites its own training code, then the training code is not necessarily an input of the AI; yet, this would still arguably be recursive self-improvement in the correct philosophical sense.)


# 4. Addressing the academic community where it's at
This paper engages with an important topic that is, to some degree, at the border of the overton window of academic discussion. As such, it is very important to address the community at a level that engages with common assumptions and the level of the discourse. Otherwise, the distance between viewpoints in the community and this paper become too large to productively engage. Just two examples that exemplify this point:
- The paper assumes some things that I find plausible, but that aren't common knowledge / commonly agreed upon in the literature. E.g., the first paragraph assumes superintelligence will be developed, without grounding this even in the literature, let alone in careful thought.
- "We can motivate this simplification by taking the long view: assuming that compute and memory keep growing exponentially, scale constraints are but a temporary obstacle." -- This clearly requires more extensive discussion or citations since there are varying viewpoints on whether compute will continue to grow exponentially in the foreseeable future.

**Questions:**

Line 102: "The fundamental property for system-internal feedback is be aligned with the external observer, and remain aligned throughout the process."

Do you mean "**to be** aligned", and do you mean "requirement" instead of "property"?

---

> ### Author Response · Authors · 2024-11-21
>
> Dear reviewer BsWX,
>
> Thank you for engaging in depth with this position paper, and for your many constructive points of feedback! We think that we can address many of them, but are happy to engage in a discussion that can bring the paper closer to the one you would have wanted to read!
>
> As you you said very accurately about a position paper like ours, it’s appropriate to:
>
> > judge its *thinking* relative to prior work: in what sense does this paper enhance the reader's ability to think clearly about issues of superintelligence and recursive self-improvement, relative to the body of literature that already exists
>
> However, we would want to clarify our intended **scope** up-front: the paper is about recursive self-improvement, yes, but *not* about superintelligence, AI safety, or the alignment problem more broadly.
>
> Also about scope, we explicitly stated (L34) that we are not aiming for this to be a survey paper (but to stimulate new thinking). That said, with currently 50+ citations drawn from a wide spectrum, the paper clearly does not live in a vacuum, with particularly dense connections to the broader landscape of research in language and RL.
>
> We think this is an appropriate scope, but from your perspective, is this narrower scope valid and useful? And deserving of ICLR visibility?
>
> > On Closedness of the system [...]
>
> There seems to be a misunderstanding here. Our aim is not to argue in favour of the closed system scenario, but to assume it and see what can be said about that more restricted setting. Never do we go so far as to claim that Socratic learning (recursive self-improvement in a closed system) is sufficient for ASI. Instead, we try to characterize the ways in which it is limited, fundamentally (and to a lesser extent, pragmatically). One key claim is that “the potential of  Socratic learning  is  high”, which challenges the potential pessimism of whether a “closed system [can] produce open-ended improvement”.
>
> We vividly agree that an "everything framework" defeats the purpose, and we have tried to be disciplined and clear about the restricted types of system we consider. The exception is the caveated statement you highlight: if this was confusing, would it sit better in a footnote?
>
>
> > The definition in the paper seems to be this sentence [...] However, the idea of recursive self-improvement is of course that the system improves itself, which is not part of this definition.
>
> The definition of self-improvement is earlier, in Section 2, which itself is using the definition of “improvement process” in Section 1. The sentence you quote zooms in on an instance of self-improvement that is also recursive.
>
> In general, we think clear definitions are extremely important for a paper like this (and we iterated them many times), so we appreciate the scrutiny!
>
> > we can consider ways of recursive self-improvement that do not feed outputs into future inputs (e.g., if the AI rewrites its own training code, then the training code is not necessarily an input of the AI; yet, this would still arguably be recursive self-improvement in the correct philosophical sense.)
>
> People use terms like “recursive” in many ways. For the purpose of our paper, we use one crisp and narrow definition, in order to make some precise follow-up statements. We discuss the broader space of related terms in its dedicated Section 6.
>
> > assumes superintelligence will be developed
>
> According to [this survey](https://arxiv.org/pdf/2401.02843), this is now a consensus view, even if timelines vary. We have added this citation.
>
> > "We can motivate this simplification by taking the long view: assuming that compute and memory keep growing exponentially, scale constraints are but a temporary obstacle."
>
> We have revised this to clarify that we need not assume exponential growth: “We motivate this simplification by taking the long view: **if** compute and memory keep growing exponentially, scale constraints are but a temporary obstacle. **If not**, considering a resource-constrained scenario for Socratic learning (akin to studying bounded rationality) may still produce valid high-level insights.
>
>
> If you have additional points that could help reach the community "where it's at", please let us know, we are happy to iterate!

---

> > ### Comment · Reviewer_BsWX · 2024-11-22
> >
> > Thank you for your rebuttal!
> >
> > > However, we would want to clarify our intended scope up-front: the paper is about recursive self-improvement, yes, but not about superintelligence, AI safety, or the alignment problem more broadly.
> >
> > I agree it is not about AI safety and alignment (though extensive discussion of these topics is still appreciated), but I am surprised that you say the text is not about superintelligence. In the abstract, you write:
> >
> > "*Considering the special case of agents with matching input and output spaces (namely, language), we argue that such pure recursive self-improvement, dubbed ‘Socratic learning,’ can boost performance vastly beyond what is present in its initial data or knowledge, and is only limited by time, as well as gradual misalignment concerns.*"
> >
> > You say performance is only limited by time and misalignment concerns. I assumed, then, that you expect such a system to scale to a misaligned superintelligence system. What is your actual view?
> >
> > > Also about scope, we explicitly stated (L34) that we are not aiming for this to be a survey paper (but to stimulate new thinking). That said, with currently 50+ citations drawn from a wide spectrum, the paper clearly does not live in a vacuum, with particularly dense connections to the broader landscape of research in language and RL.
> >
> > > We think this is an appropriate scope, but from your perspective, is this narrower scope valid and useful? And deserving of ICLR visibility?
> >
> > My concerns are less about the number of citations, and more about grounding the work in the best type of thinking about the topics that this paper addresses (recursive self-improvement, development pathways, superintelligence, limits of intelligence given training procedures). There is lots of work (with some of the best work, unfortunately, spread throughout the internet in non-archival form) that discusses these questions, and a broad engagement with it would probably allow the work to become much better e.g. by:
> > - Precisely justifying the assumptions of this work relative to prior work,
> > - Anticipating other viewpoints in readers, and,
> > - Possibly even adjusting the assumptions to better reflect reality.
> >
> > I do not change my score.
> >
> > > There seems to be a misunderstanding here. Our aim is not to argue in favour of the closed system scenario, but to assume it and see what can be said about that more restricted setting. Never do we go so far as to claim that Socratic learning (recursive self-improvement in a closed system) is sufficient for ASI. Instead, we try to characterize the ways in which it is limited [...]
> >
> > To be very frank: What is the purpose of proposing a purely philosophical framework motivated by recursive self-improvement and superintelligence if this framework can likely not realistically encompass superintelligence? I notice I am confused about what you think will be the effect of such a work.
> >
> > > We vividly agree that an "everything framework" defeats the purpose, and we have tried to be disciplined and clear about the restricted types of system we consider. The exception is the caveated statement you highlight: if this was confusing, would it sit better in a footnote?
> >
> > I don't have an opinion on this question. I could imagine that properly highlighting the purpose and exact claims/implications of the paper might help alleviate this confusion upfront.
> >
> > > People use terms like “recursive” in many ways. For the purpose of our paper, we use one crisp and narrow definition, in order to make some precise follow-up statements.
> >
> > This may be just my personal taste, but I do not understand the purpose of defining a type of recursive self-improvement that is irrelevant for what actually happens in our world. In our world, it seems very likely that the first recursively self-improving AIs will be automated ML researchers, instead of systems feeding their output back as their input. What is the take-away of considering self-improvement that likely will not manifest in the real world?
> >
> > > According to this survey, this is now a consensus view, even if timelines vary. We have added this citation.
> >
> > Thanks for adding that paper (which I was actually aware of), this is probably useful for many readers.
> >
> > ---
> >
> > I think in summary, what this paper most needs is a statement as to what readers should take away. I think a good position paper should do one of the following:
> > - Productively change the thinking of readers so as to allow them to have more accurate beliefs about the world, or:
> > - Motivate actions in the reader so as to produce more important work.
> >
> > Currently, I can't imagine this paper to have that type of influence since it seems detached from actual developments in ML and somewhat unclear in its scope regarding whether it discusses AI scaling to superintelligence.

---

> > > ### Author Response · Authors · 2024-11-25
> > >
> > > Dear reviewer BsWX,
> > >
> > > Let’s get to the core question first:
> > >
> > > > What is your actual view?
> > > > what this paper most needs is a statement as to what readers should take away
> > >
> > > Our view is quite precisely the first sentence of the abstract:
> > >
> > > **“An agent trained within a closed system can master any desired capability, as long as the following three conditions hold: [feedback, coverage, scale].”**
> > >
> > > This makes no claim either way about superintelligence. To “master any desired capability” is narrower in depth but especially in breadth – and being less ill-defined, we can say more about such learning systems. And we do: That is what the bulk of the paper is about: which requirements are needed (feedback, coverage, scale), and which restrictions are permissible (closed systems).
> > >
> > > The same interpretation also holds on the other statement you quote: “‘[Socratic learning] can boost performance vastly beyond what is present in its initial data or knowledge [+caveats].”
> > > This means what it says: impressive learning is feasible, not its extrapolation (the singularity is inevitable).
> > >
> > > Relatedly:
> > >
> > > > Productively change the thinking of readers so as to allow them to have more accurate beliefs about the world
> > >
> > > - Some readers might think: “scale is enough”, they should learn: “no, you need coverage and feedback too!”
> > > - Some readers might think: “human feedback is necessary”, they should learn: “no, not necessarily, it hinges on the reliability of the proxy: a lot can be achieved within a closed system!”
> > > - Some readers might think: “ok, recursive self-improvement works in principle, but we have no path toward it”, they should learn: “oh, maybe language games are that path!”
> > >
> > >
> > > > spread throughout the internet in non-archival form
> > >
> > > If you have specific references to point us toward, please do, but scholarly works would be more appropriate.
> > >
> > >
> > > > it seems very likely that the first recursively self-improving AIs will be automated ML researchers, instead of systems feeding their output back as their input
> > >
> > > Yes, automated AI researchers *could* well be that first instance of recursive improvement in a non-closed system – but, notably, its first manifestation in the real world (Lu et al 2024)[https://arxiv.org/pdf/2408.06292] looks very Socratic (with its self-reviewing, etc).
> > >
> > >
> > > > I do not change my score.
> > >
> > > Please do, it’s not too late! And in fact, given your argumentation skills, sway the other reviewers toward accept while you’re at it :-)

---

> > > > ### Comment · Reviewer_BsWX · 2024-11-25
> > > >
> > > > Thank you for your answer!
> > > >
> > > > On the breadth: My impression is still that the conclusion frames the contributions of the paper in terms of its applicability to create (broadly) superhuman AI, and thus I fail to find a precise limit of the scope of your ideas. To quote from the conclusion:
> > > >
> > > > > We set out to investigate how far recursive self-improvement in a closed system can take us on the path to AGI, and are now ready to conclude on an optimistic note. In principle, the potential of Socratic learning is high, and the challenges we identified (feedback and coverage) are well known.
> > > >
> > > > I do not think that you have precisely limited the scope by saying "any desired capability" in the abstract instead of "superintelligence" or "AGI". These words are not precise enough to know that you mean this in a narrower sense.
> > > >
> > > > > - Some readers might think: “scale is enough”, they should learn: “no, you need coverage and feedback too!”
> > > > > - Some readers might think: “human feedback is necessary”, they should learn: “no, not necessarily, it hinges on the reliability of the proxy: a lot can be achieved within a closed system!”
> > > > > - Some readers might think: “ok, recursive self-improvement works in principle, but we have no path toward it”, they should learn: “oh, maybe language games are that path!”
> > > >
> > > > Unfortunately, I am not aware of serious discussions of these questions in your paper. One by one:
> > > > - I think a lot of the proponents of a broad "vibe" of "scale is all you need" actually agree that you need (some form of) coverage and feedback, too. It's actually quite hard to precisely define the position that "scale will lead to AGI" in such a way that it includes the position of people like Dario Amodei but excludes those of people like Gary Marcus. I would very much love to read a serious position paper on such questions, but yours does not contribute to this question.
> > > > - On "no, it hinges on the reliability of the proxy": I am not aware of any known technique for designing a proxy that can help to achieve broadly superhuman capabilities *without* human feedback. You have not proposed a design for such a proxy, or even a paradigm for doing so. If, instead of "broadly" superhuman capabilities you just mean superhuman capabilities in *anything*, then I agree: e.g., chess computers can be designed without human feedback. But if your claim is meant so narrowly, then I don't see its novelty.
> > > > - On recursive self-improvement: In what ways are language games a way toward recursive self-improvement? You must have in mind a certain amount of breadth for this claim to become meaningful, and this breadth should be clarified.
> > > >
> > > > > If you have specific references to point us toward, please do, but scholarly works would be more appropriate.
> > > >
> > > > It is indeed a pity, but I must inform you that the most serious work on how to design superhuman AI has not appeared in scholarly (peer-reviewed) format. This leads to an awkward situation where the state of the art in current thinking is much ahead of published work, in the form of internet- and email discussions, private discussions, podcasts, and blogposts. This has two reasons: The community of people that most engaged with the topic started outside of academia, and: many ideas for how to create AGI are not publicly talked about since they are important corporate secrets. To produce novel insights, it is necessary to go beyond typical academic norms.
> > > > The most impressive work I'm aware of: [Carl Shulman's podcast with Dwarkesh](https://www.youtube.com/watch?v=_kRg-ZP1vQc).
> > > >
> > > > ---
> > > >
> > > > I might not respond directly to another round of the discussion (but might), but I will in any case take it into account in the reviewer-AC discussions afterward, should you write another answer.

---

> ### Author Response · Authors · 2024-12-03
>
> Reviewer BsWX,
>
> Thank you for your engagement and patience!
>
> On the inclusion of non-scholarly materials, we will have to agree to disagree: if the ideas are valid and important, and then someone should put them through the scrutiny of peer-review, and publish them, for everyone's benefit. The specific podcast you linked is interesting, and touches on a number of related themes (AGI, scale, self-improvement), not on others (closed systems, coverage, language games), and makes none of the same claims or arguments as our paper.
>
> > I am not aware of any known technique for designing a proxy that can help to achieve broadly superhuman capabilities without human feedback. You have not proposed a design for such a proxy, or even a paradigm for doing so. [...] In what ways are language games a way toward recursive self-improvement? You must have in mind a certain amount of breadth for this claim to become meaningful, and this breadth should be clarified.
>
> Same here, we are also not aware of any existing proxy feedback that could drive **broad** superhuman capabilities.
> But what we are aware of (and you have acknowledged as much) are a lot of proxies for *narrow* capabilities. And the central thing about language game**s** (plural) is that that may be enough: a single agent playing **many games** that each have narrow but precise and optimizable objectives could attain broad capabilities. This idea is what we tried to convey in the "if you have enough of them" section.

---

### Official Review · Reviewer_t9Dm · 2024-10-26

**Soundness:** 3
**Presentation:** 3
**Contribution:** 3
**Rating:** 5
**Confidence:** 4

**Summary:**

This position paper sets the grounds for discussion around *Socratic learning*, a concept that is used within the paper to describe recursive self-improvement in closed systems and in the language space. First, three necessary conditions for self-improvement to work are described, namely feedback, coverage and scale. The latter is assumed a non-issue in this paper so that it can focus on the other two. Next, the authors provide an explanation of what coverage and feedback conditions mean within the Socratic learning and list some of the limitations they entail for this setting. It is argued that these limitations can be addressed by the language games framework, which can effectively serve as a basis for implementing and applying Socratic learning. Finally, some recursion alternatives for self-improvement are discussed.

**Strengths:**

The discussion brought forward by this paper is a relevant one: what are the capabilities and limitations of language agents who attempt to self-improve.There is a plethora of recent works that approach this question from a technical perspective by looking at different aspects and applications such as debate or red-teaming. Coming up and proposing a comprehensible and well-structured framework which can be used for discussing this or related questions can be quite beneficial.

The paper reads very well and all its points are made clear. The transitions between the three main sections 3, 4 and 5 helped greatly the flow of the paper which ends up telling a quite coherent story. I believe this part to be particularly important for position papers.

The necessary conditions posed in this paper are something standard in the literature and hold in general for most learning settings including the one considered here. Their incorporation in the Socratic learning setting was interesting and well done, and they also helped motivating in a convincing way the use of language games within this setting. All in all, I found the statements of this paper to be reasonable, well-explained and well-grounded.

**Weaknesses:**

My main criticism for this paper is two-fold:

**Unclear contributions and insufficient covering of prior work:** The authors state in lines 33-36 that the aim of this paper is not to provide a survey of prior work, which is fair. However, even then I believe that prior/related work has not been sufficiently covered in this paper making its contributions somewhat unclear (at least to me). For example, are definitions in Section 1 all derived from prior work? And if yes where from? If not, which are novel ones and which are not? The only exception to this is recursion which is being covered properly in Section 6 of the paper. Moreover, in Section 7 you mention that all three conditions described in Section 2 are common, but you do not provide any reference in any of the two sections + I would expect this acknowledgement to already come in Section 2. Is this the first work to propose the use of language games for recursive self-improvement or related topics? I strongly believe that the paper would benefit by including a related work section discussing works that have attempted similar discussions and making clear which parts of the paper are novel and to what extent.

**Lack of concrete example:** Even though I enjoyed reading this paper and I found its claims reasonable, I missed a concrete example that connects the described methodology to real-world applications. For instance, you could include any of the recent applications that have an LLM agent self-playing with the goal of improving in some aspect, e.g., negotiation, debate, fact checking, etc.

**Questions:**

**Q1:** What *compatible* inputs and outputs exactly mean in line 138?

**Q2:** How should Socratic learning and the framework you are proposing be modified when the Observer dynamically changes, instead of being stationary as implicitly assumed by the paper?

---

> ### Author Response · Authors · 2024-11-21
>
> Dear reviewer t9Dm,
>
> Thank you for your constructive feedback, and for the many positive things you had to say about the paper! We think that the criticisms you raise can be addressed, and are eager to engage with you on how to improve the paper in these dimensions.
>
> > are definitions in Section 1 all derived from prior work? [...] If not, which are novel ones and which are not? [...] you do not provide any reference in any of the two sections
>
> The definitions are our own, and novel as far as we are aware, but of course closely related to how others are using these terms: unfortunately, as argued [here](https://arxiv.org/pdf/2407.10583), there are few canonical agent definitions. For our purposes, we wanted to have concise, self-contained definitions that permit the exposition of the remaining arguments, without imposing familiarity with additional formalisms on the reader. But we have now added in a number of additional references in those sections (Abel et al, Colas et al, Fernando et al, Ladosz et al, Lu et al, Pislar et al, Wang et al).
>
> > Is this the first work to propose the use of language games for recursive self-improvement?
>
> Yes, we think ours is a novel proposal: if you are aware of this kind of usage of language games, we’d be eager to see!
>
> > you could include [...] LLM agent self-playing with the goal of improving in some aspect, e.g., negotiation, debate, fact checking, etc.
>
> We already had pointers to LLM debate examples (Du et al, Liang et al) and included two references to negotiation papers too (Lewis et al, FAIR et al).
>
> > I missed a concrete example that connects the described methodology to real-world applications
>
> Our paper is by its nature far removed from practical applications. But we think there are very real problems that Socratic learning can be applicable to. Here is a brief example, which we’ll integrate into the next revision of the paper. Consider a math-centric system:
> - The agent reads and writes mathematical statements and proofs.
> - The observer’s performance metric is binary: did the agent find a proof for the Riemann hypothesis.
> - The system outside the agent contains a proof verifier (eg Lean), and a collection C of theorems and conjectures.
> - The proxy reward available to the agent is +1 for each verified new proof of any of the statements in collection C.
> - The system allows the agent to formulate new statements, verify its proofs about them, and add them to a second collection L (lemmas, or subgoals).
> - Over time, the agent may learn to simplify and decompose existing theorems, accumulate lemmas in L, learn to formulate lemmas that are more and more reusable, and increase the fraction of theorems for which it can produce valid proofs. At some point, the expanding frontier of verified mathematical knowledge (L plus proven statements in C) reaches the Riemann hypothesis, and the observer, satisfied, stops the system.
>
> This example of Socratic learning matches our definition, is open-ended within a closed system, could be non-trivially useful, and it seems not a priori implausible (especially since the AlphaProof results).
>
> > What compatible inputs and outputs exactly mean in line 138?
>
> Inputs and outputs are compatible when they live in the same space, and when (at least some) outputs of the agent can be fed back as inputs. For example if both are English text, or both are Python code, or both are formal math statements. As we pointed out in a footnote, input and output spaces are not necessarily identical, but they intersect. For example, the agent could be generating code, but perceive a broader set of things, like natural language, (partly self-generated) code, and execution traces.
>
> > How should Socratic learning and the framework you are proposing be modified when the Observer dynamically changes, instead of being stationary as implicitly assumed by the paper?
>
> Intriguing question! Indeed, our framing of how the observer’s performance metric is measured and can improve assumes an observer that does not change their mind about what “good” means. If the observer’s preferences are non-stationary, the problem may be ill-defined, or intractable for an agent in a closed system (as it cannot access any information about how the observer changes). It seems to us that the non-stationary scenario would be better served by an entangled setup with direct feedback, instead of a closed system.
>
> We will expand on all these thoughts in the next revision of the paper, but are happy to discuss any further questions or suggestions you may have!

---

> > ### Comment · Reviewer_t9Dm · 2024-11-25
> >
> > Dear authors,
> >
> > Thank you for your response. I have two clarification questions.
> >
> > (a) To make sure I understood correctly. You have not included a related work section, but instead additional references scattered across the paper, right? Since you have not made somehow the changes in your revision distinguishable, e.g., by highlighting them with some different color, it is difficult for me to say what exactly changed and where.
> >
> > (b) Thank you for providing a concrete example. However, you have not commented on whether the conditions you are characterizing as necessary within your paper would hold in this example, and what would this imply. I think this is a very important aspect of connecting the results and ideas from a position paper like yours to real world applications.

---

> > > ### Author Response · Authors · 2024-11-25
> > >
> > > Quick response to the revisions question: OpenReview has a built-in comparison mechanism that creates exactly the kind of diff-document you're looking for. Go to "Revisions" at the top of this page, select which versions to compare -- it seems the server is under some load today, but maybe [this link](https://openreview.net/revisions/compare?id=LsZxlxA9da&left=q3kHx5aero&right=sUAWXb9g72&pdf=true&version=2) (scroll down) works? You can also download all pdf revisions to look at them side-by-side if you prefer.
> > >
> > > In terms of what has changed at a high level: our claims are the same, but we've tried to revise for clarity, and weaved in a lot of additional references, situating this paper more solidly in the contemporary literature. The example is new too, but given its box layout that's easy to spot :-)
> > >
> > > ----
> > > ----
> > > Very good, let's look at the three requirements (feedback, coverage, scale) for the Socratic Riemann example.
> > >
> > > - feedback: as we're in a domain with formal verifiability, any proposed proof can be checked, to feedback is 100% reliable.
> > > - coverage: this is the most interesting question in this example, and there is no a priori answer. The observer cares about exactly one point in proof space (the Riemann hypothesis). The system starts from small set of proofs (not including that one) and keeps growing it. However, there are infinitely many provable statements, so the system could keep churning along productively, forever, without ever reaching the observer's goal. Note that this is not just merely a matter of efficiency of the search process, but whether it can get stuck in a narrow subspace of provable mathematics.
> > > - scale: if the coverage condition holds, then scale is still necessary. Optimistically, if an AlphaZero-like recipe could be applied, the system should be expected to iterate over millions if not billions of proofs (just like AlphaZero trained itself up on tens of millions of chess games).
> > >
> > > In other words, for this example, feedback is a given, coverage is unknown, scale is necessary.
> > >
> > > Is this a productive way of thinking about real-world problems? We think so: in this specific instance it shines a light on a likely bottleneck (coverage) that should be addressed on day one -- as opposed to heading out with a naive "scale is all you need" mindset.

---

> > > > ### Comment · Reviewer_t9Dm · 2024-11-25
> > > >
> > > > Ah thank you for letting me know about the comparison mechanism, I admit I was unaware of that.
> > > >
> > > > Thank you also for describing the conditions. I believe explicitly mentioning them to your example would improve clarity.
> > > >
> > > > (b contd) Does the fact that coverage might not be satisfied in this example means that recursive self-improvement might just not lead to the agent learning the desired proving skill you describe? If yes, shouldn't this then imply that your Socratic learning approach might not be effective in this example, bc the conditions under which it would be are simply impractical?
> > > >
> > > > (a contd) Your rebuttal has not lifted my main concern regarding coverage and comparison of related work. It is less about the number of citations and more about how your definitions and proposed approach connects and compares to similar or related ideas and concepts within your framework. For example, you mentioned yourselves in your rebuttal, that there are other existing agent definitions. However, in the paper you do not draw a comparison with them, or you simply just don't elaborate on why the one you are proposing should be the one adopted. Now agents here is just an example, the same applies to other concepts within your framework (see my original review for more details). Also a lack of related work section makes it difficult to assess how novel your definitions of the concepts within your framework are, and hence the size of your contributions I will repeat becomes blurry bc of this.

---

> > > > > ### Comment · Reviewer_t9Dm · 2024-12-02
> > > > >
> > > > > I have read the revised manuscript. Since my main concerns about this paper were not addressed during the rebuttal, I will maintain my negative score.
> > > > >
> > > > > Ps. To shape a more well-informed opinion, I also read the discussions between authors and other reviewers. I found the response of the authors to Reviewer BsWX, on whether they will increase their score or not, at least inappropriate. I am not writing this to defend my fellow reviewer, it is not my place to do so, but I would like to strongly advise the authors to be more respectful in how they respond in rebuttals in the future.

---

> ### Author Response · Authors · 2024-12-03
>
> Dear reviewer t9Dm,
>
> In your original review you stated two concerns. The "lack of a concrete example" has been very directly addressed. The "insufficient covering of prior work" is more debatable, but we have made significant changes in this direction, as well as clarified that not all the terms we define have standard comparable definitions elsewhere. For example about the agent definition we responded that, no, there are not many canonical ones:
>
> > The definitions are our own, and novel as far as we are aware, but of course closely related to how others are using these terms: unfortunately, [as argued here, there are few canonical agent definitions](https://arxiv.org/pdf/2407.10583). For our purposes, we wanted to have concise, self-contained definitions that permit the exposition of the remaining arguments, without imposing familiarity with additional formalisms on the reader. But we have now added in a number of additional references in those sections (Abel et al, Colas et al, Fernando et al, Ladosz et al, Lu et al, Pislar et al, Wang et al).
>
> Also, to be clear, while some of our definitions are novel to the best of our knowledge, we do not claim that the definitions are a contribution in and of themselves: they are only spelled out to make the arguments of the paper as clear and self-contained as possible.
>
> But as always, if you have specific pointers to related definitions or important comparisons, please tell us and we will happily integrate them!
>
>
> > (b contd) Does the fact that coverage might not be satisfied in this example means that recursive self-improvement might just not lead to the agent learning the desired proving skill you describe? If yes, shouldn't this then imply that your Socratic learning approach might not be effective in this example, bc the conditions under which it would be are simply impractical?
>
> Yes, exactly, *if* the coverage condition is not satisfied, then the system will not reach its goal. Socratic learning *can* be effective, in this example and in general, when the three conditions hold. For some systems we may be able to make strong statements about those, and hence predict success. For other systems, we may need to fall back on empirical evidence. But even in the empirical scenario, awareness of the three conditions can benefit progress, as compared to only hill-climbing on the external success metric.
>
> > I found the response [...] at least inappropriate
>
> We have high respect for the time our reviewers put into this process, and have aimed to respond in an open, collaborative spirit. We believe it is possible to be both direct and respectful (even if a deferential style is common elsewhere).

---

### Official Review · Reviewer_zfgR · 2024-10-27

**Soundness:** 2
**Presentation:** 3
**Contribution:** 2
**Rating:** 6
**Confidence:** 2

**Summary:**

The article discusses a type of learning in which the learning agent is self-reflective, rather than relying on external data and knowledge inputs to improve itself, as is typical of most existing learning agents. In fact, even the most advanced large language models (LLMs) today strengthen their capabilities through data-driven training and human feedback. However, the author suggests an alternative: a learning agent that is capable of introspective or self-improving learning within a closed environment—improving itself without external inputs. This self-contained learning, termed “Socratic learning,” is feasible if the environment meets three conditions: (a) sufficiently informative and aligned feedback, (b) broad coverage of experience or data, and (c) adequate capacity and resources, the latter of which the author believes is only a matter of time.

The author presents a unique framework called “language games” as an implementation example to support Socratic learning, explaining how multi-player dynamics and the design of language games into many narrow but well-defined language games—rather than a single, universal one—may satisfy the requirement for broad coverage. Additionally, features like a meta critic could make these language games scalable. The author also suggests feasible approaches to ensure feedback consistency.

**Strengths:**

The article takes a higher-level view of today’s artificial intelligence, introducing "Socratic learning" as a potential path toward AGI and offering an implementation framework for realizing Socratic learning. The ideas presented are thought-provoking and stimulate valuable reflection.

**Weaknesses:**

This article is intended to be a thought-provoking piece, aimed at stimulating reflection and discussion. Since this is a conceptual position, its suitability for a technical conference like ICLR may require further evaluation.

**Questions:**

As a reviewer, I have one observation. If I’ve understood correctly, Socratic learning presupposes that the game’s agent/player is already somewhat intelligent, meaning it can independently derive insights, learn from experiences, and comprehend new knowledge without external input. However, today’s agents, even the most advanced LLMs, lack true intelligence; they generate responses probabilistically rather than through logical understanding. Therefore, should the design of an intelligent agent itself also be considered a prerequisite for Socratic learning?

---

> ### Author Response · Authors · 2024-11-20
>
> Dear reviewer zfgR,
>
> Thank you for your positive and constructive review; it is great to see that our ideas got you thinking, hopefully into new directions!
>
> > its suitability for a technical conference like ICLR may require further evaluation
>
> We acknowledge this is a reason for being cautious, but ICLR can accept position papers (and has in the past, as helpfully pointed out by reviewer BsWX), but maybe we can leave that dimension to the AC, and in the meantime work on the substance of the paper?
>
> > Socratic learning presupposes that the game’s agent/player is already somewhat intelligent, meaning it can independently derive insights, learn from experiences, and comprehend new knowledge without external input. However, today’s agents, even the most advanced LLMs, lack true intelligence; they generate responses probabilistically rather than through logical understanding. Therefore, should the design of an intelligent agent itself also be considered a prerequisite for Socratic learning?
>
> This is a very insightful suggestion! Indeed, there are conditions on the agent: most obviously, the agent must be capable of learning (otherwise its performance cannot improve). And it must have the capacity to learn a lot, potentially, depending on how demanding the observer is. In broad terms, we included this under the umbrella term of “scale” in section 2.3, and the questions of “which algorithms work” (L125).
>
> While capability to learn is definitely a prerequisite, your second point about “the design of an intelligent agent” is more nuanced: the last decade has produced a lot of evidence that learning from scratch, with zero initial competence, can lead to excellent performance (eg DQN, AlphaZero). But for language domains, the situation is less clear-cut: learning to communicate, let alone full language, is challenging to do from scratch (eg [here](https://proceedings.neurips.cc/paper/2019/hash/fe5e7cb609bdbe6d62449d61849c38b0-Abstract.html)), and even when it does not start from scratch language can drift away from anything humans can understand (eg [here](https://arxiv.org/abs/1706.05125)). Hence the temptation to use LLMs as very strong starting points for intelligent agents. We agree with you that LLMs are not sufficient unless they keep learning, and would even venture that it is not obvious that they can learn enough, nor that the community knows how to train them in an open-ended way. But a research program that targets ambitious Socratic learning (instead of pleasing chatbot users) may by necessity lead to the development of such algorithms.
>
> We will expand on all these thoughts in the next revision of the paper, but are happy to discuss any further questions or suggestions you may have!

---

> > ### Author Response · Authors · 2024-11-25
> >
> > Dear reviewer zfgR,
> >
> > We have now revised the paper, with your input and that of the other reviewers: if you have time to have another look before the discussion period closes, that would be great!

---

> > > ### Comment · Reviewer_zfgR · 2024-11-25
> > >
> > > Thank you for your response. I can accept categorizing efficient learning under the scaling option, as it does seem like a solvable issue given a sufficiently long timeline, which arguably renders it trivial.
> > >
> > > In the first round of review, I rated myself highly due to my familiarity with the Reinforcement Learning field. However, after reading other reviewers' comments, I realized that I am not actually very familiar with language games. This has led me to believe that I should lower my self-assessment score in this area.
> > >
> > > I will keep my score for the paper unchanged for now. That said, as this is a position paper with fair contributions, I don’t have a particularly strong opinion about whether it should be accepted at ICLR.

---

### Official Review · Reviewer_3DHg · 2024-10-28

**Soundness:** 2
**Presentation:** 2
**Contribution:** 2
**Rating:** 3
**Confidence:** 5

**Summary:**

This is a position paper that argues for open-ended learning in `a closed system' based on the framework of 'language games.'
The agent interacts with itself using natural language inputs and outputs like a group of Socratic philosophers talking to themselves. The question raised is if this kind of interaction can result in self-improvement ultimately leading to 'artificial super intelligence.'

**Strengths:**

This is a thought-provoking position paper and not a technical paper. It raises some interesting questions and makes connection to frameworks such as reinforcement learning and language games. The idea of reaching super-human intelligence through self-improvement is a tantalizing goal, but leads to many questions I outline in the next section.

**Weaknesses:**

Unfortunately the claims of the paper are not compelling. I could appreciate the analogy to reinforcement learning, where the agent is trying to optimize a given reward function. However in the absence of such as a known reward function or human feedback, it is unclear how the "self-improvement" is supposed to occur.

"External to the system is an observer whose purpose is to assess the performance of the agent"

However, if the agent is oblivious to this performance assessment, how does it help the agent improve it? If the agent does receive feedback and adapt itself to improve its performance, this seems identical to RL with human feedback.

The idea of language games is an old one, but again it is not clear what the goal of such a game is supposed to be, and what guarantees 'alignment' of meaning or purpose with humans. Communication between people (and philosophers) is possible due to shared meaning, shared life experiences, and shared culture.  Not sure how a pure language model that interacts with itself can improve itself.

Some concrete examples of the concepts put forth here would have been more useful.

Consider for example a mathematical language like first order calculus. A theorem prover might start with something like  Peano axioms and derive a lot of new facts about integers. Or someone can axiomatize chess and derive some interesting consequences. Does it qualify as an example of what you have in mind? These systems have some inherent limitations in that what they can learn is limited to what follows from the axioms. Further, it is known that the proof systems are incomplete and intractable even in the simplest of cases.

In summary, this is a thought-provoking position paper. Despite the authors' attempts to clarify their paper and answer the reviewers' probing questions, I remain unconvinced about the novelty and significance of the arguments.

**Questions:**

Please give some concrete examples of the framework you have in mind. Argue how such a system can be constrained to reach a desirable goal or if it is open-ended how it can be guaranteed to be safe.

---

> ### Author Response · Authors · 2024-11-20
>
> Dear reviewer 3DHg,
>
> Thank you for your thoughtful and constructive feedback. For a moment, let us suspend the question of what is the right venue for such a position paper (and maybe leave that to the AC –  in any case, ICLR can accept position papers, and has in the past, as pointed out by reviewer BsWX), and engage on its substance:
>
> > in the absence of such as a known reward function
>
> Yes this precisely what we highlight as one of the two key challenges to recursive self-improvement, in section 2.1. We emphasise the critical importance of setting the system up with a feedback proxy (introduced in L48) that is sufficiently aligned: as you state correctly, without it the agent cannot improve. One type of proxy is a classic (programmatic) reward function coded up by the observer/designer, in which case the system internally does “just” RL, and the agent may or may not learn the behaviour that was intended (cf the large literature on reward hacking). Another form of proxy could be a formal verifier (like the Lean prover in AlphaProof) that can provide grounded feedback on the agent’s progress.
>
> In section 5, we discuss another mechanism for feedback: not one language game, but many language games, each endowed with a scoring function that can drive the agent’s learning. On top of this space of language games sits then another less easily grounded “meta-game” that informs which games the agent plays (based on what playing them and learning from them produces in terms of validation performance, for example). Some of this is already revised for clarity in the latest version (using Carse’s notion of finite and infinite games), but we will iterate further on this.
>
> > or human feedback
>
> Our emphasis on a closed system deliberately excludes this kind of learning from the discussion, and zooms in on what is doable *without* human feedback. In no way do we want to say that human feedback is not useful, or not desirable, but we think the question of what can be attained without it is an interesting one.
>
> > Some concrete examples of the concepts put forth here would have been more useful.
>
> Luckily, as this is a non-technical paper, it is not too late for that (and the paper is not currently at its page limit), so we will add one or two worked examples to the next revision.
>
> For a brief example right now, consider a math-centric system:
> - The agent reads and writes mathematical statements and proofs.
> - The observer’s performance metric is binary: did the agent find a proof for the Riemann hypothesis.
> - The system outside the agent contains a proof verifier (eg Lean), and a collection C of theorems and conjectures.
> - The proxy reward available to the agent is +1 for each verified new proof of any of the statements in collection C (and 0 otherwise).
> - The system allows the agent to formulate new statements, verify its proofs about them, and add them to a second collection L (lemmas, or subgoals).
> - Over time, the agent may learn to simplify and decompose existing theorems, accumulate lemmas in L, learn to formulate lemmas that are more and more reusable, and increase the fraction of theorems for which it can produce valid proofs. At some point, the expanding frontier of verified mathematical knowledge (L plus proven statements in C) reaches the Riemann hypothesis, and the observer, satisfied, stops the system.
>
> This example of Socratic learning matches our definition, is open-ended within a closed system, safe and aligned, useful and non-trivial, and it seems not a priori implausible (especially since the AlphaProof results).
>
> Are there more questions that the paper has raised that you’d like to spell out or discuss?

---

> > ### Comment · Reviewer_3DHg · 2024-11-21
> >
> > Thank you for your response. The example adds some concreteness to the discussion.
> >
> > However, what you described is what typically happens in a theorem prover. There are statements proved, subgoals to be proved, and a way to check that the proofs are correct. Not that theorem proving is a solved problem, but it is not clear what your formulation adds to the state of the art. As you say, Alphaproof already exists.

---

> > > ### Author Response · Authors · 2024-11-22
> > >
> > > In your original review, you asked:
> > >
> > > > Not sure how a pure language model that interacts with itself can improve itself. Some concrete examples of the concepts put forth here would have been more useful.
> > >
> > > Our example tried to address this, by giving a plausible illustration of how an agent (which is not identical to, but may contain a language model) can indeed improve itself within a closed system. But it seems you have now changed your mind and no longer wonder how Socratic learning could be possible, but instead think it is possible, but trivially so?
> > >
> > > But that’s fine, let’s work with that, and tease out a few more differences.
> > >
> > > First, there are clear differences between a classical theorem prover and AlphaProof: while both search over proofs they can verify, the latter also includes learning (once it finds a proof to a problem X, its search for proofs of related problems is sped up). On the results side, its proofs for IMO problems are far beyond what classical provers could have found.
> > >
> > > In the terminology of our paper, a classical theorem prover is an agent, but a rather simple one whose performance remains fixed over time, so not a learning agent (and hence not self-improving either). The details of AlphaProof remain unpublished, so it is difficult to be certain, but it seems like the closest entity to our notion of agent is their “solver network”, which is trained to get better and better at searching for proof steps, using RL. If we assume that this solver network takes past proof steps (which it generated itself) as inputs, then its inputs and outputs are “compatible” in our paper’s terminology, and that makes it a recursive self-improvement system, or almost: The missing property is that of the system being closed. For that to be true, new problems cannot be added from the outside, but be generated by the agent itself. This seems doable in principle, just not what they actually did (see also [this recent work](https://arxiv.org/pdf/2407.00695) that pairs a solver with a conjecturing component).
> > >
> > > That said, even if there were no differences, and AlphaProof was an exact instance of Socratic learning, that should be a boon to our paper, not a disadvantage? If a successful instance exists already, that should strengthen the case for taking the framework seriously?
> > >
> > > So now we can address this concern:
> > >
> > > > it is not clear what your formulation adds to the state of the art
> > >
> > > Mathematics is one of the clearest domains for an illustration, but it is merely one example domain. In fact, compared to the full scope of the paper, it is a bit too simple: with its formal verifiability it sidesteps the difficulty of feedback (Section 2.1). For maths, the primary challenges lie in coverage and scale (Sections 2.2 and 2.3) instead. In contrast, other domains like debate, theory of mind, or other language games, may struggle less with coverage and more with feedback.
> > >
> > > Our formulation permits the reader to think in one unified way about many such settings, with some extra clarity arising from the imposed boundaries of the closed system. We have been cautious to not overclaim the degree of novelty, but our framework and formalism is distinct from any prior work and (we think) sheds fresh light on an important topic.
> > >
> > > Finally, we’d like to ask you, respectfully but bluntly: you have rather rated this paper as 1 out of 10 – why? What is your primary concern? It is a position paper whose purpose is to provoke thought and raise questions, it is an opening, an invitation, not a closing statement. On these terms: are the raised questions interesting questions? Yours and the other reviewers’ questions seem non-trivial and interesting to us, we have answered them, and already the paper has benefitted from that interaction. May this not be a paper that benefits the ICLR community?

---

> > > > ### Comment · Reviewer_3DHg · 2024-11-25
> > > >
> > > > Thank you for your patient responses to all the reviews.
> > > >
> > > > Yes, my primary concern about your paper was that it was a position paper. I don't think it is fair to compare it against technical papers. Perhaps there should have been a separate category and criteria for acceptance of position papers in the conference.
> > > >
> > > > I increased my score taking this into account. However, I still fail to see the novelty of the contribution if theorem proving (and search?) can be viewed as a valid instantiation of your framework. As I said the paper contains some thought-provoking ideas,
> > > > but to me a position paper must point out something original that is missed by the community and needs attention. I am sorry that I am not able to see that here.

---

> ### Author Response · Authors · 2024-12-03
>
> Dear reviewer 3DHg,
>
> > I still fail to see the novelty of the contribution if theorem proving (and search?) can be viewed as a valid instantiation of your framework. As I said the paper contains some thought-provoking ideas, but to me a position paper must point out something original that is missed by the community and needs attention.
>
> Whether an idea "needs attention" is inherently somewhat subjective, but if you consider that recursive self-improvement could be or soon become a driver of progress in the community, then ideas closely related to that should be of high interest?
>
> Do we state things that have been "missed by the community"? Unambiguously, yes. We are not aware of any prior work (nor has any been pointed out in this review process) that discusses the limitations of learning within closed systems specifically, or that proposes to use language games for the purpose of recursive self-improvement.
>
>
> Also, from our earlier response to reviewer BsWX:
>
> >> - Some readers might think: “scale is enough”, they should learn: “no, you need coverage and feedback too!”
> >> - Some readers might think: “human feedback is necessary”, they should learn: “no, not necessarily, it hinges on the reliability of the proxy: a lot can be achieved within a closed system!”
> >> - Some readers might think: “ok, recursive self-improvement works in principle, but we have no path toward it”, they should learn: “oh, maybe language games are that path!”

---

### Official Review · Reviewer_8rsu · 2024-11-04

**Soundness:** 2
**Presentation:** 2
**Contribution:** 2
**Rating:** 3
**Confidence:** 2

**Summary:**

This paper explores the concept of Socratic learning, a recursive self-improvement process in the context of language model.  Central to this approach is the framework of language games, derived from Wittgenstein's philosophy, which facilitate interactive data generation and provide feedback mechanisms through scoring. The paper seems more proper to a philosophical or language conference, as the current manuscript consists no quantitive evaluation or concrete algorithm presentation, as noted by the authors.

**Strengths:**

Idea is somewhat interesting by drawing connection of learning and philosophical concepts.

**Weaknesses:**

1. No concrete algorithm for the proposed framework;
2. No quantitive evaluation of the proposed framework

**Questions:**

How would the proposed framework and language game cope with any existing language model (e.g., llama, ChatGPT)? Can the authors perform some prototyping and validation of the proposed framework?

---

> ### Author Response · Authors · 2024-11-20
>
> Dear reviewer 8rsu,
>
> it seems that your primary concern with the paper is that it is a position paper, not a technical paper. Please note that ICLR can accept position papers (and has in the past, as pointed out by reviewer BsWX).
>
> Engaging with it on its own terms: Do you think the ideas we propose are novel, useful, precise, “thought-provoking” as the other reviewers state? Do you think the position is an interesting one, and that our arguments are valid? If not, could you provide constructive feedback? If yes, would you mind adjusting your score?

---

> ### Comment · Reviewer_8rsu · 2024-11-27
> **Official Comment by Reviewer 8rsu**
>
> Dear Authors,
>
> Thank you for your thoughtful response. After careful consideration, I have decided to maintain my original score for this paper. While the work is valuable, I did not find it particularly "thought-provoking" or feel that it addresses a specific gap that is currently missed by the community when evaluated as a position paper.

---

### Author Response · Authors · 2024-11-20

Dear reviewers, thank you for engaging so constructively with this paper: We hope that together with you we can refine this into a valuable and stimulating read for the whole ICLR community!

First off, we are grateful for your encouraging words about how “thought-provoking” the paper is, appreciating that the “important topic” is “underdiscussed”, and pursuing a “tantalizing goal”. And on the execution, we’re pleased to hear you think it is “interesting and well done”, “well-explained and well-grounded”. We think the highest praise is that we’re “at the border of the Overton window of academic discussion”, in which case: yes, let’s open that window a bit further and let some fresh air in!

We’re keenly aware this is an atypical paper for you to review. Maybe the AC can weigh in on this, but in principle ICLR can accept position papers (and has in the past, as pointed out by reviewer BsWX).

We will fix issues of clarity, provide a concrete example, tie in related work, tighten up our arguments: we’ll upload a revised draft soon, and respond to your specific concerns individually.

Given all this, it seems worthwhile to discuss and iterate to get the paper to a clear accept! Again, thank you for putting in the time to consider this paper and meet it on its own terms.

---

> ### Author Response · Authors · 2024-11-22
> **Revised draft**
>
> We have now updated a draft with numerous improvements suggested by the reviewers. The three main changes are improvements to clarity throughout, a worked example of Socratic learning in the domain of maths (end of Section 3), and ~20 additional references to related literature. We're happy to address another round of comments, if needed!

---

> ### Author Response · Authors · 2024-12-03
>
> P.S: Just for fun, [here](https://notebooklm.google.com/notebook/89b7764a-df15-4a9c-b503-25fb07f1f5f4/audio?pli=1) is an AI-generated podcast about the paper that we just came across.

---

### Meta-Review · Area_Chair_ZKdZ · 2024-12-19

**Metareview:**

The reviewers acknowledged that the paper explores an interesting question of how recursive self-improvement could lead to mastering any desired capabilities, and the paper provides several thought-provoking ideas in this direction. However, the reviewers pointed out several weaknesses and shared common concerns related to the unclear scope of the paper, missing engagement with prior work, and lack of concrete examples. We want to thank the authors for their detailed responses. The paper was discussed extensively by the reviewers, and there was a consensus that the paper is not ready for acceptance. Based on the raised concerns and follow-up discussions, unfortunately, the final decision is a rejection. Nevertheless, the reviewers have provided detailed and constructive feedback. We hope the authors can incorporate this feedback when preparing future revisions of the paper.

**Additional Comments On Reviewer Discussion:**

The reviewers pointed out several weaknesses and shared common concerns related to the unclear scope of the paper, missing engagement with prior work, and lack of concrete examples. The paper was discussed extensively by the reviewers, and there was a consensus that the paper is not ready for acceptance.

---

### Decision · Program_Chairs · 2025-01-22

Reject